# Seasonal Pattern of Cerebrovascular Fatalities in Cancer Patients

**DOI:** 10.3390/healthcare11040456

**Published:** 2023-02-04

**Authors:** Velizar Shivarov, Hristo Shivarov, Angel Yordanov

**Affiliations:** 1Department of Experimental Research, Medical University Pleven, 5800 Pleven, Bulgaria; 2Singing River Hospital, Pascagoula, MS 39567, USA; 3Department of Gynaecological Oncology, Medical University Pleven, 5800 Pleven, Bulgaria

**Keywords:** cerebrovascular disease, death, cancer, season, cosinor model, SEER program

## Abstract

Cancer patients are at increased risk of cerebrovascular events. The incidence of those events and the associated mortality are known to follow a seasonal pattern in the general population. However, it is unclear whether cerebrovascular mortality in cancer patients has seasonal variation. To address this question, we performed a retrospective analysis of the seasonality of deaths due to the fact of cerebrovascular diseases among patients with first primary malignancy registered between 1975 and 2016 in the SEER database. The presence of seasonality in death rates was modeled using the cosinor approach assuming a circa-annual pattern. A significant seasonal pattern with a peak in the first half of November was identified in all patient groups. The same peak was observed in almost all subgroups of patients defined based on demographic characteristics. However, not all entity-defined subgroups showed a seasonal pattern, which might be explained by the different pathologic processes affecting the circulatory system in each cancer type. Based on our findings, one can propose that the active monitoring of cancer patients for cerebrovascular events from the late autumn and during the winter can help in the reduction of mortality in this patient population.

## 1. Background

In 2018, cerebrovascular diseases accounted for 37.1 deaths per 100,000 US citizens making them the fifth leading cause of death in that population (https://www.statista.com/topics/4385/cerebrovascular-disease-in-the-us/#topicHeader__wrapper (accessed on 29 December 2022). On the other hand, during the same period the second most common cause of death in the US general population was cancer (1). Recent reports confirmed that cancer patients are at increased risk for cerebrovascular diseases and associated deaths [1,2,3]. A plethora of factors contribute to the increased risk of cerebrovascular diseases, including direct tumor effects and cancer-associated coagulopathies and infections, as well as the direct and indirect consequences of chemo-, radio, and supportive therapy and invasive diagnostic and therapeutic interventions [4]. Interestingly, seasonal variations in physiological processes governed by environmental stimuli, such as variations in the daylight duration, ambient temperature, and humidity, can contribute to the seasonal variations in the incidence of health-related events, such as morbidity and mortality from communicable and noncommunicable diseases. We recently analyzed systematically the effect of cancer diagnosis on the seasonal variation of deaths due to the fact of suicidal behavior and heart diseases. We demonstrated that suicidal and heart-related deaths in cancer patients follow robust seasonal patterns similar to the general population suggesting that cancer diagnosis does not eliminate the effect of environmental stimuli, which drive the pathophysiological process [5,6]. In this regard, the seasonal variation in the cerebrovascular disease presentation and mortality have been widely studied over the last five decades, with the majority of studies demonstrating detectable peaks in morbidity and mortality in the cold months of the year [7,8,9,10,11,12,13,14]. However, to the best of our knowledge, there has been no report regarding the seasonality of cerebrovascular deaths among cancer patients. Therefore, we aimed to investigate whether cerebrovascular mortality in cancer patients showed any type of seasonality and whether it was affected by additional demographic- or cancer-related factors.

## 2. Materials and Methods

### 2.1. Data Availability

We retrieved US cancer patient records with the documented cause of death from the Surveillance, Epidemiology, and End Results (SEER) program diagnosed between 1975 and 2016 from 18 registries (November 2018 submission), as described previously [5,6]. We retrieved a total of 106,512 unique cases, which had a documented cause of death recode of “cerebrovascular disease”. For the purposes of this analysis, we focused only on the entries of first malignant primary if a patient had been diagnosed with several primary malignancies. The SEER database does not provide data regarding the exact date of death of each patient. Therefore, we estimated the month of death for each patient included in the analysis based on the available month and year of diagnosis and survival time in months. For the purposes of this estimation, we assumed an average duration of a month of 30.25 days and the day of diagnosis as the day of the reported month of diagnosis. In order to classify counties of residence of cancer patients into southern and northern counties subgroups, we retrieved the geospatial data for US counties from the US Census Bureau website https://www.census.gov/geographies/reference-files/time-series/geo/gazetteer-files.html (accessed on 29 December 2022). Classification into southern and northern counties based on the median of the latitudes of the counties included in the study, which was 37°55′10″ N. The counties of residence at the time of cancer diagnosis were also classified into either metropolitan or nonmetropolitan subgroups based on their rural–urban classification in 2003 (https://seer.cancer.gov/seerstat/variables/countyattribs/ruralurban.html (accessed on 29 December 2022). As described above, this study made use only of publicly available anonymized data from a cancer registry and from US authorities and, therefore, it was considered exempt from ethics committee review, and no procedure for obtaining informed consent was considered applicable.

### 2.2. Statistical Analysis

We recently demonstrated the use of a classical cosinor model with one cycle per year to demonstrate the presence of seasonality for suicidal deaths and deaths from heart diseases among cancer patients [5,6]. Based on this experience, we decided to apply identically the same approach to cerebrovascular deaths among cancer patients, as described below. In brief, the cosinor model is summarized by the following equation:(1)$$f\left(t\right)=Acos\left(\frac{2\pi t}{c}-P\right).$$

In this formula, *A* denotes the amplitude of the sinusoidal curve, and *p* is used to denote its phase, whereas c is the length of the seasonal cycle (in our case *c* = 12 for 1 cycle per year) and *t* means the time of each observation [15]. This equation can be linearized to the following:(2)$$Y_{t}=c\cos\left(\omega_{t}\right)+s\sin\left(\omega_{t}\right),\;t=1,\ldots ,\;n.$$

This transformation allows for the calculation of the amplitude and the phase using the estimates *c* and *s* from the equation above [15]. In order to maintain the overall significance level of the model be maintained at α = 0.05, the assumed levels of significance for the cosine and sine terms were set to less than 0.025 [15]. All analyses were performed using R v. 4.2.1 for Windows (64-bit) and the package *season* (v. 0.3.15) [16]. All presented plots were generated using the package *ggpubr* (v. 0.5.0).

## 3. Results

We identified a total of 106,512 patients with primary malignancies and the term for the documented cause of death “Cerebrovascular Diseases” reported in the SEER database between 1975 and 2016 (based on 18 registries’ data submitted in November 2018). The demographic features of the patients included in the analysis are represented in Table 1, and expectedly they were identical to the overall distribution reported previously [1,2]. We derived the date of death by adding the reported survival time to the month of diagnosis, as described above. The median of the estimated age at death was 83 years, with a range between 0 and 114 years (Table 1).

We initially fitted the cosinor model to all patient cohorts of identified patients with death due to the fact of cerebrovascular disease. The model demonstrated a significant seasonal pattern with a peak in the first half of November (Table 2 and Figure 1). We further fitted the cosinor model to subgroups defined based on gender, age at death, race, time of death since cancer diagnosis, and latitude of the county of residence. All models with the exception of younger patients (under 50 years) and race other than Black and White showed a significant peak in October–December (Table 2 and Figure 1). The patients with death due to the fact of cerebrovascular disease in the first year after the diagnosis of primary malignancy had a peak in early December (Table 2). Both patients residing in metropolitan and nonmetropolitan counties at the time of diagnosis had the typical peak in cerebrovascular mortality in early November (Table 2 and Figure 1). Additionally, irrespective of the period of diagnosis, the peak was in the first half of November. The exception from that was the subgroup of patients diagnosed between 2010 and 2016 for which an earlier peak in the second half of September was estimated. The obvious explanation for this outlier is that the data cut-off was in November 2018 and, therefore, not all death cases for this group were captured in the SEER database.

The patients included in the analysis had different underlying malignancy suggesting different treatment modalities, which could potentially affect the risk of cerebrovascular disease. We defined 16 subgroups of patients by entity and fitted the cosinor model to each of them in order to identify a single cycle circa-annual pattern [5,6]. The groups that had a significant seasonality at an alpha level of 0.025 were the patients with lymphoma and myeloma, breast cancer, lung and bronchus cancer, colorectal cancer, head and neck cancer, female genital cancer, prostate, urinary cancer, and skin melanoma (Table 2 and Figure 2). Most of the entity-defined subgroups had a peak in November, but there were several groups with earlier peaks (Table 2). For example, patients with skin melanoma and lung and bronchus cancer had peaks in late September. On the other hand, patients with lymphoma or myeloma and head and neck cancer had peaks of death due to the fact of cerebrovascular diseases in October.

## 4. Discussion

Cerebrovascular disease and especially stroke are still considered a global health issue with a significant increase in stroke-related mortality over the last three decades [17]. The main contributing risk factor for this increase has been the increase in the body mass index (BMI) [17]. However, acute cerebrovascular (as well as cardiovascular events in general) are commonly precipitated by exacerbations in pathophysiological factors, such as hypertensive crises, conditions associated with hypercoagulability, and acute infections [18]. Indeed data from the Global Burden of Disease Study showed that the main risk factors for stroke worldwide between 1990 and 2019 were high systolic blood, pressure high body mass index, high fasting plasma glucose, ambient particulate matter pollution, and smoking [19]. The first four of those are directly or indirectly dependent on seasonal variations in ambient conditions and season. Therefore, it might not be considered surprising that acute cerebrovascular morbidity and mortality can have seasonal pattern. Indeed, a large number of studies addressed this assumption. Studies from the Northern Hemisphere showed peaks of hospitalizations and deaths from acute cerebrovascular events in autumn and winter [20,21]. For example, more than four decades ago Haberman et al. [7] showed that the peak of deaths from cerebrovascular disease in England and Wales for both males and females was in March. There was a small difference in the peaks between both sexes when cerebrovascular diseases were analyzed separately [7]. Males had peak mortality from cerebrovascular hemorrhage and thromboembolic events in January, whereas in females the respective peaks were observed in March [7]. These data were confirmed for other European countries [22]. However, to the best of our knowledge, the question of whether cerebrovascular diseases have seasonal variation in incidence and mortality remains unexplored. In addition, some studies suggest that cancer patients may have additional distinct mechanisms for cerebrovascular diseases in comparison to the general population [3,23]. Based on those reports, we aimed to explore whether cerebrovascular deaths in cancer patients follow any seasonal pattern. To address this question, we used data from 106,512 unique cases of cerebrovascular disease-related fatalities in all age groups in cancer patients diagnosed between 1975 and 2016 (Table 1). As described by us previously, we fitted a standard cosinor model assuming a 12 month annual cycle for all patient groups and various subgroups defined either by demographic characteristics or by the primary entity [5,6].

We identified a significant peak in cerebrovascular mortality in early November for the entire group. This peak is consistent with the peak we recently reported for deaths from cardiovascular diseases in cancer patients from the SEER database suggesting that the driver mechanisms of cardiovascular and cerebrovascular mortality in cancer patients are identical [5]. Notably, the peak in cardiovascular (including cerebrovascular diseases) mortality in the general population in USA is during winter (January and February) [24]. Specific studies on defined cerebrovascular diseases such as ischemic stroke showed peak mortality in the general population in winter [25]. Previous large studies from the USA also showed that peak mortality from stroke in the USA was observed during the winter, which was identical to the peak in deaths due to the fact of myocardial infarction [26,27]. However, hospitalizations for acute stroke may have a different seasonal pattern in the USA [28] and other regions [29,30].

One can speculate that our approach was biased toward the identification of an earlier peak, as we estimated the months of death based on the months of diagnosis and survival in months. This bias, however, would be insignificant, and the maximum shift in the peak if primary data were used would not be later than early December. In addition, studies from other countries also showed a peak in stroke mortality in the general population in autumn [20]. In any case, it is obvious that the peak in cerebrovascular mortality in cancer patients might be earlier than the one in the general population. This may be a true phenomenon with a clear mechanistic explanation, as the vascular system of cancer patients may be more sensitive to the short-term effect of abrupt change in some environmental factors, such as ambient temperature [18,31,32,33,34] and air pollution [35] or be a consequence of the seasonal variation in physiological processes, such as coagulation and inflammatory response [36,37,38]. These possible explanations could not be explored in more detail because of the main obvious limitation of this study in that the cause of death “Cerebrovascular Diseases” as reported in the SEER database comprises several entities and specific cancer types that might be prone to experiencing death from one of these entities rather than from others. For example, hemorrhagic stroke might be more frequent in patients with acute leukemia during the first year after diagnosis compared to other more indolent malignancies requiring less aggressive therapy. To partially address this issue, we analyzed seasonality in cerebrovascular deaths during the first year from diagnosis as well as per groups of malignancies. Notably, both early and late deaths from cerebrovascular diseases showed significant peaks; however, the peak in early deaths (within one year after cancer diagnosis) was in early December, whereas the peak for deaths after the first year was in early November. It is unclear whether this is entirely due to the fact of climatic factors or due to the fact of cancer intrinsic of therapy-related factors. In support of the idea that climatic factors (such as ambient temperature) were not the most important precipitating factor for cerebrovascular diseases in cancer patients was the fact that patients from both northern and southern counties had identical peaks in deaths in early November. Interestingly, when we analyzed the presence of a seasonal pattern per entity not all subgroups showed a significant seasonal pattern. For example, leukemia patients did not show a significant peak in cerebrovascular mortality, whereas the lymphomas and myelomas subgroup did. Furthermore, among the subgroups with significant seasonality, the time for the model to peak varied. Patients with melanoma and lung and bronchus cancer showed a peak in late September, and patients with myelomas and lymphomas had a peak in October. The most plausible explanation for these differences, as mentioned above, would be different patterns of cerebrovascular diseases in different cancer subtypes, which are influenced by the different therapy modalities being used. The contribution of the latter is also not directly addressable because of the scarce and inconsistent reporting of therapy for the patients included in the SEER database. Surprisingly, the peaks observed over the five decades covered in the study were consistent (Table 2). This suggests that the changes in the management of different entities did not interfere with the seasonal pathophysiological mechanisms triggering cerebrovascular mortality over a relatively long period of time. Finally, access to medical care is unlikely to affect seasonality in cerebrovascular deaths among cancer patients because patients residing in either metropolitan or nonmetropolitan counties had identical November peaks (Table 2).

## 5. Conclusions

Taken collectively, our analyses suggest that the incidence of cerebrovascular disease-related fatalities after cancer diagnosis follows a seasonal pattern similar to the one observed for the general population. The peak incidence of cerebrovascular disease-related deaths is probably earlier than in the general population, which may be due to the alteration in the circulatory system associated with cancer therapy, making it more sensitive to physiological changes triggered by the bioclimatic factors in late autumn. This peak is consistent with the peak identified for mortality from heart disease-related deaths in cancer patients. Finally, the obvious practical implication of our findings to the fields of oncology and neurology is that cancer patients must be monitored closely for cerebrovascular events from late autumn and during the entire winter period, which may potentially lead to reduced mortality in this population.

## Figures and Tables

**Figure 1 healthcare-11-00456-f001:**
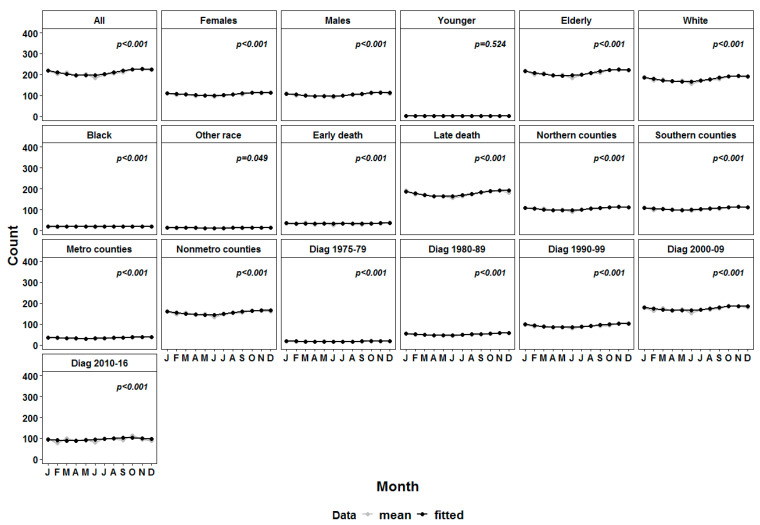
Curves of the number of deaths due to the fact of cerebrovascular diseases in all patients and main subgroups (in grey) versus the fitted cosinor model values (in black).

**Figure 2 healthcare-11-00456-f002:**
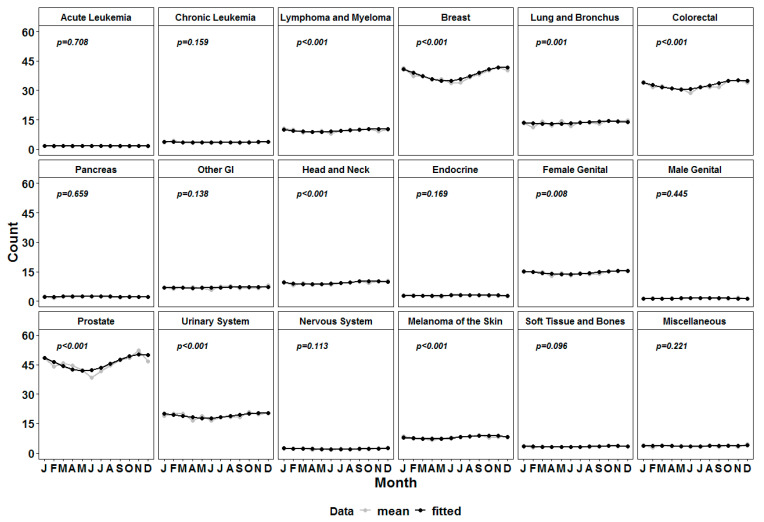
Curves of the number of deaths due to the fact of cerebrovascular diseases per month in the main cancers subtypes (in grey) versus the fitted cosinor model values (in black).

**Table 1 healthcare-11-00456-t001:** Demographic characteristics of the included cancer patients with death due to the fact of cerebrovascular disease.

	Numbers
Total	10,6512
Sex	
Male	52,754
Female	53,758
Age at diagnosis	
Median	74
Range	(0–107)
Age at death	83
Range	(0–114)
Race	
White	90,534
Black	9711
Other	6030
Unknown	237
Year of diagnosis	
1975–79	10,483
1980–89	22,474
1990–99	29,830
2000–2009	35,688
2010–16	8037
Counties	
Northern	53,311
Southern	54,209
Metropolitan	17,001
Nonmetropolitan	77,844

**Table 2 healthcare-11-00456-t002:** Parameters of the cosinor model fits to all cancer patients with death from cerebrovascular disease and defined subgroups.

Group	Count	Amplitude	Peak Month	Lowest Month	*p*-Value Cosine Term	*p*-Value Sine Term
Per Demographic Characteristic						
All patients	106,512	16.093	11.1	5.1	3.23845 × 10^−19^	6.5648 × 10^−47^
Males	52,754	8.988	11	5	1.04818 × 10^−10^	9.3789 × 10^−32^
Females	53,758	7.144	11.2	5.2	5.18766 × 10^−10^	6.1423 × 10^−18^
Younger	1122	0.084	1.8	7.8	0.523940865	0.77733231
Elderly	105,390	16.033	11	5	9.46695 × 10^−19^	1.5508 × 10^−47^
White	90,534	14.397	11.1	5.1	2.7511 × 10^−19^	5.8411 × 10^−43^
Black	9711	1.136	10.4	4.4	0.432691032	0.00011406
Other race	6267	0.613	11.7	5.7	0.048754141	0.11459755
Northern counties	53,201	8.477	11.1	5.1	3.99012 × 10^−12^	6.5836 × 10^−26^
Southern counties	53,311	7.624	11	5	9.76055 × 10^−9^	9.5916 × 10^−23^
Early death	17,198	1.653	12.1	6.1	0.000104147	0.04056075
Late death	89,314	15.147	11.1	5.1	5.35391 × 10^−22^	1.0853 × 10^−46^
Metro counties	17,001	3.447790714	10.9	4.9	3.27 × 10^−5^	1.94 × 10^−15^
Nonmetrocounties	77,844	11.61872517	11.1	5.1	2.44 × 10^−14^	8.52 × 10^−34^
Diag 1975–1979	8378	1.824073317	11.5	5.5	4.62 × 10^−6^	3.05 × 10^−6^
Diag 1980–1989	22,474	5.388099508	11.3	5.3	6.68 × 10^−11^	3.55 × 10^−17^
Diag 1990–1999	29,830	9.692679699	11.2	5.2	1.71 × 10^−13^	3.86 × 10^−22^
Diag 2000–2009	35,688	10.73202203	11	5	0.000109922	3.58 × 10^−12^
Diag 2010–2016	8037	6.309932231	9.3	3.3	0.153544163	0.00015054
Per Entity						
Acute leukemia	405	0.044	6.7	12.7	0.707513445	0.95305883
Chronic leukemia	1590	0.186	12.6	6.6	0.159100828	0.77956333
Lymphoma and myeloma	4645	0.778	10.7	4.7	0.186547703	0.0004353
Breast	19,130	3.656	11.5	5.5	3.55294 × 10^−10^	1.7356 × 10^−10^
Lung and bronchus	6798	0.770	9.9	3.9	0.855507401	0.0013389
Colorectal	16,519	2.407	11.1	5.1	0.000299859	9.6657 × 10^−8^
Pancreas	865	0.069	5.5	11.5	0.662814512	0.65917269
Other GI	3384	0.256	10	4	0.993301983	0.1378723
Head and neck	4606	0.921	10.3	4.3	0.459219169	9.4038 × 10^−6^
Endocrine	968	0.230	8.2	2.2	0.168909257	0.33371912
Female genital	7251	0.892	11.6	5.6	0.008110391	0.01637148
Male genital	259	0.101	7.6	1.6	0.445206048	0.82498158
Prostate	23,190	4.462	11.2	5.2	1.58859 × 10^−9^	1.9769 × 10^−15^
Urinary system	9606	1.429	11.5	5.5	0.000429642	0.00038066
Nervous system	767	0.198	12.5	6.5	0.112611056	0.64141945
Melanoma of the skin	3459	0.813	9.7	3.7	0.53388816	6.2737 × 10^−5^
Soft tissue and bones	1316	0.248	10.8	4.8	0.469157981	0.09566596
Miscellaneous	1754	0.187	11.9	5.9	0.221414287	0.42869463

## Data Availability

Source data are publicly available through the Surveillance, Epidemiology, and End Results (SEER) Program.

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
