# Peer review of "Seasonal Pattern of Cerebrovascular Fatalities in Cancer Patients"

_healthcare, 2023, doi:10.3390/healthcare11040456_

Round 1
Reviewer 1 Report
This is retrospective analysis of seasonality of deaths due to cerebro-vascular diseases among patients with first primary malignancy registered between 1975 and 2016 in the SEER database. A significant seasonal pattern with a peak in the first half of November was identified in the all patients group. The same peak was observed in almost all subgroups of patients defined based on demographic characteristics. However, not all entity-defined subgroups showed seasonal pattern, which might be explained by the different pathologic processes affecting circulatory system in each cancer type. Based on authors findings one can propose that active monitoring of cancer patients for cerebrovascular events from the late autumn and during the winter can help reduce mortality in this patient population.
The study is well written and adds new knowledge to the topic of cerebro-vascular diseases. Nevertheless, there is no reference to current publications regarding risk factors for death in patients with ischemic stroke, e.g. Xu et al, Wańkowicz et al.
Author Response
Dear Sir,
Thank you for your comments. We appreciate your suggestion to include a reference regarding the risk factors for stroke. In order to be more comprehensive we cited the latest publication from the GBD study regarding the risk factors for stroke. We added the following text:
Indeed data from the Global Burden of Disease Study showed that the main risk factors for stroke worldwide between 1990 and 2019 were e high systolic blood, pressure high bodymass index, high fasting plasma glucose, ambient particulate matter pollution and smoking [19]. The first four of those are directly or indirectly dependent on seasonal variations in ambient conditions and season.
Thanks,
Velizar Shivarov
Reviewer 2 Report
I read with great interest the paper authored by Velizar Shivarov about seasonal pattern of cerebrovascular fatalities. The topic is within the scope of the Healthcare, MDPI. Overall, the study is easy to follow and adds new findings to the current literature.
Only one comment:
1. description of race as a other more than 6000 cases. What about the Asian race?
2. Data was retrieved from the USA patient's record. What is more correct for this data: Black or African American ?
Author Response
Dear Sir/Madam,
Thank you for reviewing our manuscript. To answer your comments:
- This is the term reported in the SEER database. We followed their terminology.
- Asian origin was not reported independently in SEER for the period targeted in our study. Therefore, we were not able to analyze that subpopulation separately.
Sincerely,
Velizar Shivarov